

# Alteration of RNA m6A methylation mediates aberrant RNA binding protein expression and alternative splicing in condyloma acuminatum

Xiaoyan Liu[1,*], Bo Xie[2,*], Su Wang[1], Yinhua Wu[1], Yu Zhang[1] and Liming Ruan[1]

[1] Department of Dermatology, the First Affiliated Hospital, Zhejiang University School of Medicine, Hangzhou, Zhejiang, China
[2] Department of Urology, the First Affiliated Hospital, Zhejiang University School of Medicine, Hangzhou, Zhejiang, China
[*] These authors contributed equally to this work.

Corresponding authors
Xiaoyan Liu, hzliuxiaoyan@zju.edu.cn
Liming Ruan, 1188021@zju.edu.cn

## ABSTRACT

**Background.** Condyloma acuminatum (CA) is caused by low-risk human papillomavirus, and is characterized by high recurrence after treatment. The RNA modification N6-methyladenosine (m6A) plays an important role during diverse viral infections, including high-risk HPV infection in cervical cancer. However, it is unclear whether low-risk HPV infection changes the RNA m6A methylation in CA.

**Methods.** High-throughput m6A-sequencing was performed to profile the transcriptome-wide mRNA modifications of CA tissues infected by LR-HPVs and the paired normal tissues from CA patients. We further investigated the regulation of alternative splicing by RNA binding proteins (RBPs) with altered m6A modification and constructed a regulatory network among these RBPs, regulated alternative splicing events (RASEs) and regulated alternative splicing genes (RASGs) in CA.

**Results.** The results show that the m6A level in CA tissues differed from that in the paired controls. Furthermore, cell cycle- and cell adhesion- associated genes with m6A modification were differentially expressed in CA tissues compared to the paired controls. In particular, seven RNA binding protein genes with specific m6A methylated sites, showed a higher or lower expression at the mRNA level in CA tissues than in the paired normal tissues. In addition, these differentially expressed RNA binding protein genes would regulate the alternative splicing pattern of apoptotic process genes in CA tissue.

**Conclusions.** Our study reveals a sophisticated m6A modification profile in CA tissue that affects the response of host cells to HPV infection, and provides cues for the further exploration of the roles of m6A and the development of a novel treatment strategy for CA.

## INTRODUTION

Condyloma acuminatum (CA) is a benign growth that often occurs in the genitals and anus and is usually caused by human papillomavirus (HPV) infection mainly through sexual transmission. More than 90% of cases of CA are caused by low-risk (LR) HPV (*Zhou et al., 2019*; *Lu et al., 2020*), while high-risk (HR) HPV infection is closely related to cervical cancers and oropharyngeal cancers (*Rahangdale et al., 2022*). Notably researchers pay much attention to the pathogenesis of HR-HPV infection in cancer, which reveals mechanisms of the interaction between HPVs and oncogenes or tumor suppressor genes (*Oyouni, 2023*). CA is characterized by high recurrence after treatment (*Park, Introcaso & Dunne, 2015*), which leads to the thorny problem of controlling the spread of this disease. Therefore, it is of great importance to identify the underlying pathogenesis of CA.

The RNA modification N6-methyladenosine (m6A), the most widespread and dynamic posttranscriptional modification, functions by affecting a series of processes, such as RNA transcription, processing, transportation, translation and degradation (*Wang et al., 2020*), which are involved in the regulation of biological processes such as the occurrence and development of cell differentiation diseases (*Jiang et al., 2021*). In fact, m6A is also involved in regulating the host's immune response to viral infection (*McFadden & Horner, 2021*). On the one hand, m6A on the viral genome allows the viral genome to disguise itself as host RNA and evade the recognition of nonself by the host (*Lu et al., 2020*; *Lu et al., 2021b*); on the other hand, infection of viruses leads to changes in the m6A expression of host cellular RNA (*Liu et al., 2019*; *Gokhale et al., 2020*; *Kim et al., 2021*; *Liu et al., 2021*; *Statello et al., 2021*), which is involved in diverse cellular processes (*Liu et al., 2019*) and innate immune activation (*Gokhale et al., 2020*; *Kim et al., 2021*). Recently, it was reported that HR-HPV 16/18 infection in cervical cancer stabilized m6A-methylated MYC expression by regulating IGF2BP2, an m6A reader, further promoting aerobic glycolysis (*Hu et al., 2022*). Increased m6A modification of interferon-$\epsilon$ dependent on WT1-associated protein was revealed to mediate the antiviral response in CA (*Gu et al., 2023*). However, there are still very few data on transcriptome m6A changes and their probable functions and regulatory mechanisms in the pathogenesis of CA.

M6A modification changes the charge, conformation, expression and RNA binding protein (RBP) anchoring of modified RNA (*Su et al., 2021*); sometimes it plays its vital roles dependent on RBPs with "reader" or "nonreader" functions (*Chen et al., 2019*) to control nearly all aspects of mRNA processing, including alternative splicing (AS), mRNA trafficking, stability and translation (*Qin et al., 2020*). Proline-rich coiled-coil 2B, as both a m6A reader and an RBP, regulates AS of the alpha 1 chain of collagen type XII in a m6A-dependent manner to mediate hypoxia-induced endothelial cell migration (*Li et al., 2023*). NF-kappa B activating protein, an RBPs, promotes cystine-glutamate antiporter mRNA splicing by binding to m6A and further suppressing ferroptosis (*Sun et al., 2022*).

AS is also a kind of posttranscriptional regulatory mechanism in eukaryotes that increases protein diversity by processing pre-mRNA into mRNA isomers. This mechanism is also widely involved in virus infection. It was reported that hepatitis delta virus (HDV) induced AS changes in host genes through splicing factor SF3B155 sequestration and contributed

to the early progression to hepatocellular carcinoma in HDV-infected patients (*Tavanez et al., 2020*). HSV-1 is also likely to restrict the expression of randomly activated antigenic viral genes to escape from host immune recognition by splicing machinery (*Tang, Patel & Krause, 2019*). For HPV, HR-HPVs express their proper genes by extensive use of mRNA AS mediated by cellular RBPs (*Kajitani & Schwartz, 2022*). However, the AS profiles in CA tissues caused by LR-HPV and their probable generative mechanism remain unknown.

To address the above question, we used m6A-seq technology to profile the transcriptome-wide mRNA modifications of CA tissues infected by LR-HPVs and the paired normal tissues from CA patients. We further investigated the regulation of AS by RBPs with altered m6A modification and constructed a regulatory network among these RBPs, regulated alternative splicing events (RASEs) and regulated alternative splicing genes (RASGs) in CA. By demonstrating the relationship between m6A-methylated RBPs and dysregulation of AS in CA, the present study provides supplemental information to help unravel the probable molecular and cellular mechanisms of CA pathogenesis.

## METHODS

### Patients and specimens

Approval for this study was obtained from the Institutional Review Board of the First Affiliated Hospital, School of Medicine of Zhejiang University (2022-795). Written informed consent was obtained from all subjects. The flow of the procedural steps in our study is shown in Fig. 1. CA female patients aged 20-40 years old with positive HPV genotyping were included, and those complicated with tumors, other immunocompromised diseases and other sexually transmitted diseases were excluded. A total of five pairs of HPV6 or 11-positive CA specimens (CA group) and their adjacent mucosal tissues (CON group) were collected for m6A immunoprecipitation and m6A sequencing. Furthermore, an additional 40 HPV-positive CA specimens (CA group) and 15 HPV-negative vaginal mucosal tissues (CON group) were collected for further verification using quantitative real-time PCR (RT-qPCR). The clinical characteristics of CA patients were shown in Table S1. All samples were removed using surgical scissors and were stored in liquid nitrogen immediately and kept in liquid nitrogen for not more than 1 month for further detection. Swab samples (*Sarier et al., 2023*) were obtained from the warts before excision to get the exfoliated cells, and DNA were delivered for HPV genotyping using a 23 HPV Genotyping Real-time PCR Kit (Hybribo, China), including 17 high-risk HPV types of HPV 16, 18, 31, 33, 35, 39, 45, 51, 52, 53, 56, 58, 59, 66, 68, 73, 82, and six low-risk HPV types of HPV 6, 11, 42, 43, 44, and 81.

### M6A immunoprecipitation and m6A sequencing

Mature polyadenylated mRNAs were isolated from total RNA according to the GenElute™ mRNA Miniprep Kit (Sigma, Burlington, MA, USA) instructions and fragmented to a length of 100 nt using RNA fragmentation buffer. Then, m6A enrichment was performed by incubating Pierce™ ChIP-grade Protein A/G Magnetic Beads (26162; Thermo Fisher Scientific, Waltham, MA, USA) washed with IP buffer with 12.5 μg of anti-m6A antibody (202003; Synaptic Systems, Germany). Then the m6A antibody-binding RNA was eluted

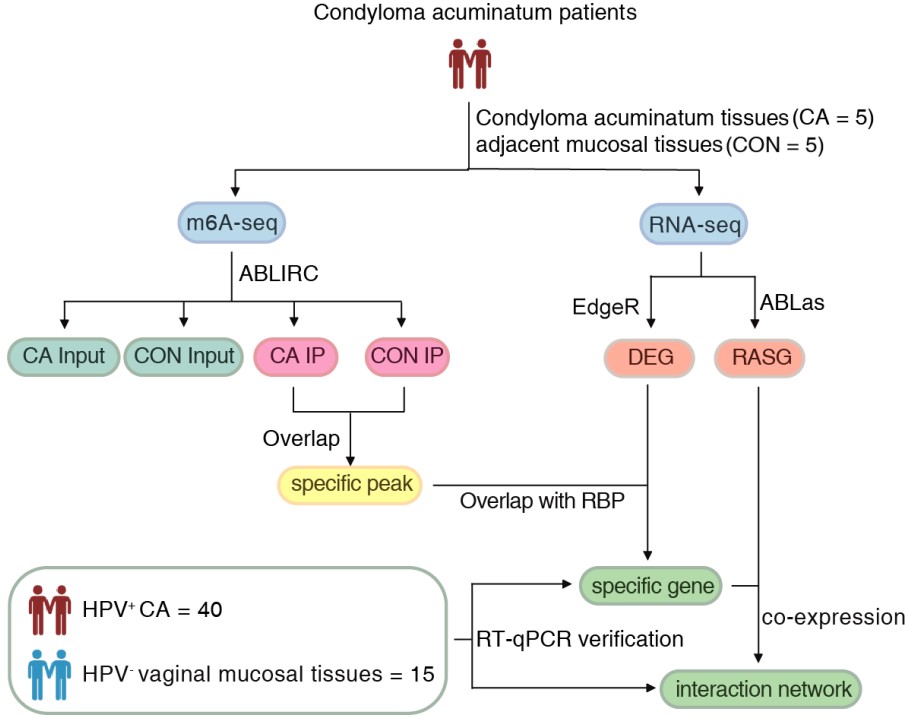

**Figure 1** **The flow of the procedural steps in the present study.**

from the magnetic beads using m6A elution buffer, and the supernatant was collected for purification. Finally, the IP and Input samples were reversely transcribed to prepare cDNA libraries according to the protocol of the RNA-Seq Library Preparation Kit, and the quality-checked libraries were then subjected to PE150 sequencing using the Illumina HiSeq 2000 platform at LC-Bio Biotech, Ltd. (Hangzhou, China). The raw sequence data were deposited in the Gene Expression Omnibus (GEO) database under the accession number GSE172140.

## Analysis of m6A-seq data

Cutadapt was used to remove adaptors and low-quality bases from raw reads, and reads less than 16 nt were discarded to obtain high-quality clean reads. The number of reads covered by the IP libraries was significantly higher than that of the Input, which could result in "peaks" and be used to indicate the methylation loci in the genome. The final differential peaks were called and screened through the ABLIRC pipeline (*Xia et al., 2017*). For each peak, the unique number of reads, the lengths of reads, and the observed maximum peak height were counted. Meanwhile, the computer simulations were used to generate reads with the same number and length for each gene. These outputting reads were mapped onto the same genes, to generate random maximum peak heights from overlapping reads. Finally, all observed peaks whose height is higher than the random maximum peak with a $P < 0.05$ were selected. In addition, Homer software is used to show the shape of m6A peak.

### Differentially expressed gene analysis

Clean reads were aligned to the human GRch38 genome by Tophat2 (*Kim et al., 2013*), allowing four mismatches. Based on the uniquely mapped reads, the fragments per kilobase of transcript per million fragments mapped (FPKM) for each gene were calculated to evaluate the gene expression levels. Differentially expressed genes (DEGs) were identified using edgeR software (*Robinson, McCarthy & Smyth, 2010*) and screened with a cutoff of FC ($\geq 2$ or $\leq 0.5$) and FDR <0.05 as the thresholds to determine whether genes were differentially expressed.

### Alternative splicing analysis

Alternative splicing analyses (ASEs) and RASEs between samples were defined and quantified by using the ABLas pipeline (*Jin et al., 2017*; *Xia et al., 2017*). Briefly, ABLas detected 10 major splicing types based on splice junction reads, including exon skipping (ES), alternative 5′ splicing site (A5SS), alternative 3′ splicing site (A3SS), intron retention (IR), mutually exclusive exons (MXE), mutually exclusive 5′ UTRs (5pMXE), mutually exclusive 3′ UTRs (3pMXE), box exons, A3SS&ES and A5SS&ES. To clarify the RBP-regulated ASEs, Student's $t$-test was used to calculate the statistical $P$ value after ASE detection for each sample, and the RASE ratio was defined based on the change in alternative reads and model reads of samples, whereas those with a significant change in splicing ratio and $P$ <0.05 were considered to be significant ASEs regulated by RBP.

### Coexpression analysis

The coexpression regulation patterns between RBP expression and RASEs were determined by calculating their Pearson correlation coefficients (PCCs) and corrected $P$ values, and could be further classified as positive, negative and noncorrelation based on the PCC values.

### Functional enrichment analysis

Functional enrichment of GO terms and KEGG pathways for all DEGs was performed using the KOBAS 2.0 server (*Xie et al., 2011*). The hypergeometric test and Benjamini−Hochberg FDR controlling procedure were used to define the enrichment of each term. Reactome pathway analysis was also performed online using the Reactome database (http://reactome.org).

### Expression validation of DEGs and RASEs by RT-qPCR

RT-qPCR was conducted on an additional 40 HPV-positive CA specimens (CA group) and 15 HPV-negative vaginal mucosal tissues (CON group) to elucidate the validity of the RNA-seq data. Primer sequences of target DEGs are shown in Table S2 and were synthesized by RiboBio (Guangzhou, China). GAPDH was used as a reference gene. The PCR conditions were as follows: denaturing for 10 min at 95 °C, then 40 cycles including denaturation for 15 s at 95 °C, with annealing and extension for 1 min at 60 °C. Three technique replicates were performed for PCR amplification of each sample.

In addition, RT-qPCR assays were applied to analyze ASEs. A boundary-spanning primer and an opposing primer in a constitutive exon were used for the sequence encompassing

the junction of constitutive or alternative exons, and detecting alternative isoforms, respectively. The boundary-spanning primer of alternative exons was designed according to "model exon" to detect model splicing or according to "altered exon" to detect altered splicing. The primer sequences for detecting ASE of RASAL2 and YWHAZ are also included in Table S2.

## Statistical analysis

The R package factoextra was used for PCA, and the samples were clustered using the first two principal components. Next-generation sequencing data and genome annotations were visualized using in-house scripts (sogen) after normalization of the reads by FPKM for each gene. Hierarchical clustering based on Euclidean distance was performed using the pheatmap package. Student's t test was used to compare AS ratio changes for target gene splicing events. Other statistical results were obtained by R software, where a $P$ value <0.05 was considered statistically significant.

# RESULTS

## Distribution characteristics and functional analysis of m6A modification in CA and paired adjacent normal tissues

A total of 20 libraries consisting of five replicates for both input and IP samples in the CA and CON groups were sequenced (Table S3). The principal component analysis (PCA) with five pairs of IP and input samples after normalizing all gene expression levels is shown in Fig. 2A. The expression level of all genes clearly distinguished the samples in the four groups were clustered. Meanwhile, the heatmap clustering analysis of sample correlation was performed based on the normalized mapped reads on each gene, showing immunoprecipitated samples clustered (Fig. S1A). Most of input and IP reads of each sample were mapped to introns, CDS, 3′UTR, and intergenic regions by analysis of the distribution of unique mapped reads (Fig. S1B). When compared with the enrichment in input controls, m6A-IP reads were enriched around both the translation start and stop codons in both CA and CON samples (Fig. 2B). Notably, the increase in CON IP samples was much more pronounced than that in CA IP samples (Fig. 2B). This results indicated that HPV infection decreased the m6A level around the translation start and stop codons of mRNA in CA.

Then, we identified m6A peaks from m6A-seq data by using the ABLIRC pipeline with input reads as backgrounds (Li et al., 2019). Peaks detected in at least three CA or CON samples were retained, which resulted in 33,885 and 12,778 m6A peaks for CA and CON, respectively (Fig. S1C). A plot of the distribution of m6A peaks showed a preferable location in the intron region (54.7%) (Fig. S1D). A de novo motif search was performed by using the HOMER algorithm for the merged m6A peaks of CON or CA samples. Motif analysis showed that both the CON and CA samples had the canonical GGACU motif of human cells in the top third motif (Figs. S1E–S1F).

Concerning the function of CA- or CON-specific m6A peak-associated genes, GO analysis revealed that CON-specific m6A peak-associated genes were enriched in terms of DNA-dependent regulation of transcription and intracellular signal transduction (Fig.

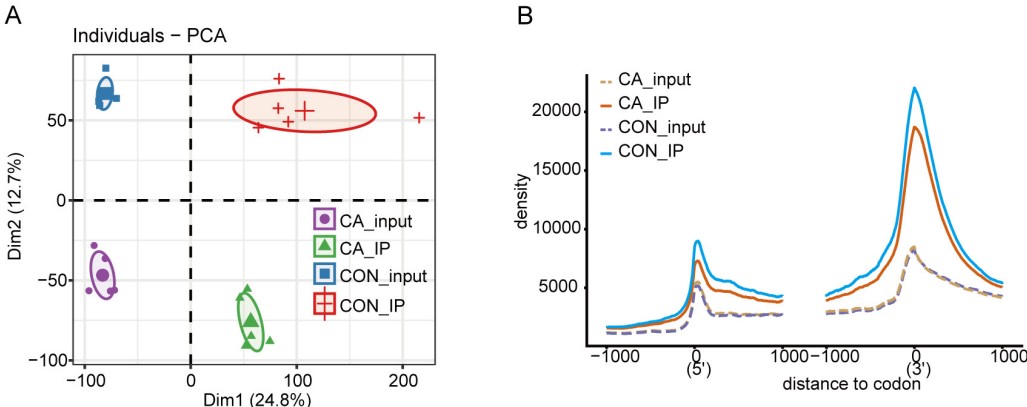

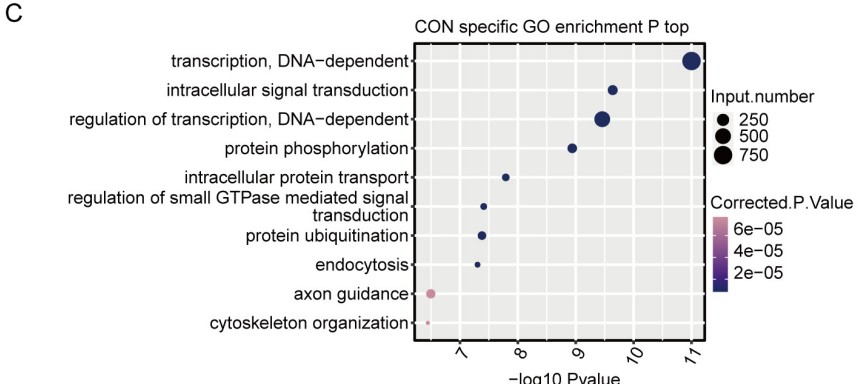

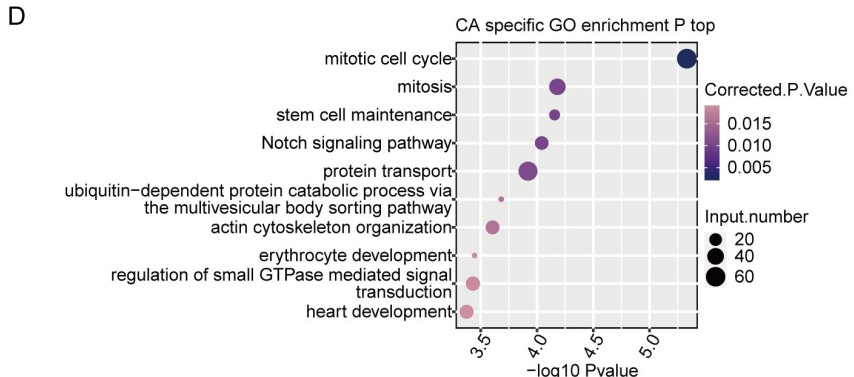

**Figure 2  Principal component analysis (PCA) and functional analysis of m6A in condyloma acuminatum (CA) and the paired adjacent normal tissues (CON).** (A) Principal component analysis (PCA) showed the sample correlation of IP and input samples. Five biological replicates were prepared for each group, and the confidence ellipse for each group is shown as an ellipse with a different color. (B) Enrichment analysis of m6A-IP and input reads around translation start and stop codons. Reads counts were normalized in a ±1000 bp window around translation start and stop codons. (C) The top ten enriched GO terms of identified CON-specific m6A peak-associated genes. (D) The top ten enriched GO terms of identified CA-specific m6A peak-associated genes.

2C), while CA-specific m6A peak-associated genes were enriched in terms of mitotic cell cycle, mitosis, stem cell maintenance and Notch signaling pathway (Fig. 2D). These results indicated that alteration of the m6A modification caused by HPV infection could participate in the regulation on cell cycle-associated genes to influence the cell proliferation or apoptosis of HPV-infected cells.

## Differentially expressed genes with m6A modification between CA and paired adjacent normal tissues

The DEGs between CA and CON samples were further analyzed to explore the potential functional consequence of m6A alteration in the RNA. PCA for these ten samples showed that CA and CON samples were clearly separated (Fig. 3A), indicating that the gene expression profile was different between CA and CON samples.

To identify the DEGs between the CA and CON groups, edgeR was used with a cutoff of fold change $\geq 2$ or $\leq 0.5$ and a 5% false discovery rate (FDR). In total, compared to CON, there were 1,519 upregulated or 1,791 downregulated DEGs in CA tissues (Fig. 3B). Hierarchical clustering of normalized FPKM values of DEGs showed a clear separation between the CA and CON groups and a high consistency for the five replicate datasets (Fig. 3C), indicating that HPV infection significantly altered the transcription expression level in CA tissue.

Gene Ontology (GO) functional analysis was performed to annotate all 3,257 DEGs to explore the potential roles of DEGs. The upregulated genes in CA tissues were significantly enriched in terms of keratinization, mitotic cell cycle, inflammatory response, chemotaxis and cell adhesion, et al. (Fig. 3D). The downregulated genes were significantly enriched in terms of extracellular matrix organization, cell adhesion and innate immune response (Fig. 3E). It was suggested that HPV infection affects the expression of genes associated with the cell cycle and immune response in CA tissues.

Furthermore, CON- or CA-specific m6A peak-associated genes overlapped with the DEGs to reveal whether m6A alteration changes the expression of genes. It was shown that 626 CA- and 1030 CON-specific m6A peak-associated genes were significantly differentially expressed in CA tissues from CON tissues (Figs. 3F and 3H). Then, further GO analysis for DEGs with CON- or CA-specific m6A peaks uncovered that DEGs with CA-specific m6A peaks were also enriched in terms inducing the cell cycle, response to cytokines and cell adhesion (Fig. 3G), and the DEGs with CON-specific m6A peaks were enriched in terms inducing cell adhesion, angiogenesis and regulation of apoptotic processes (Fig. 3I). These results indicated that HPV infection would alter the expression of m6A-modified genes functioning in the cell cycle, cell adhesion and cellular apoptosis.

## Alteration of the m6A level of RNA binding protein genes affects their expression in CA

RBPs play key roles in the regulation of HPV gene expression in HR-HPV-induced intracervical neoplasia and cervical cancer (*Kajitani & Schwartz, 2022*). Thus, we focused on CON- or CA-specific m6A peak-associated RBP genes that were differentially expressed between CA and CON samples. First, a list of 1,550 known RBPs was created from two previous studies (*Baltz et al., 2012*; *Castello et al., 2012*). Overlap analysis showed that 49
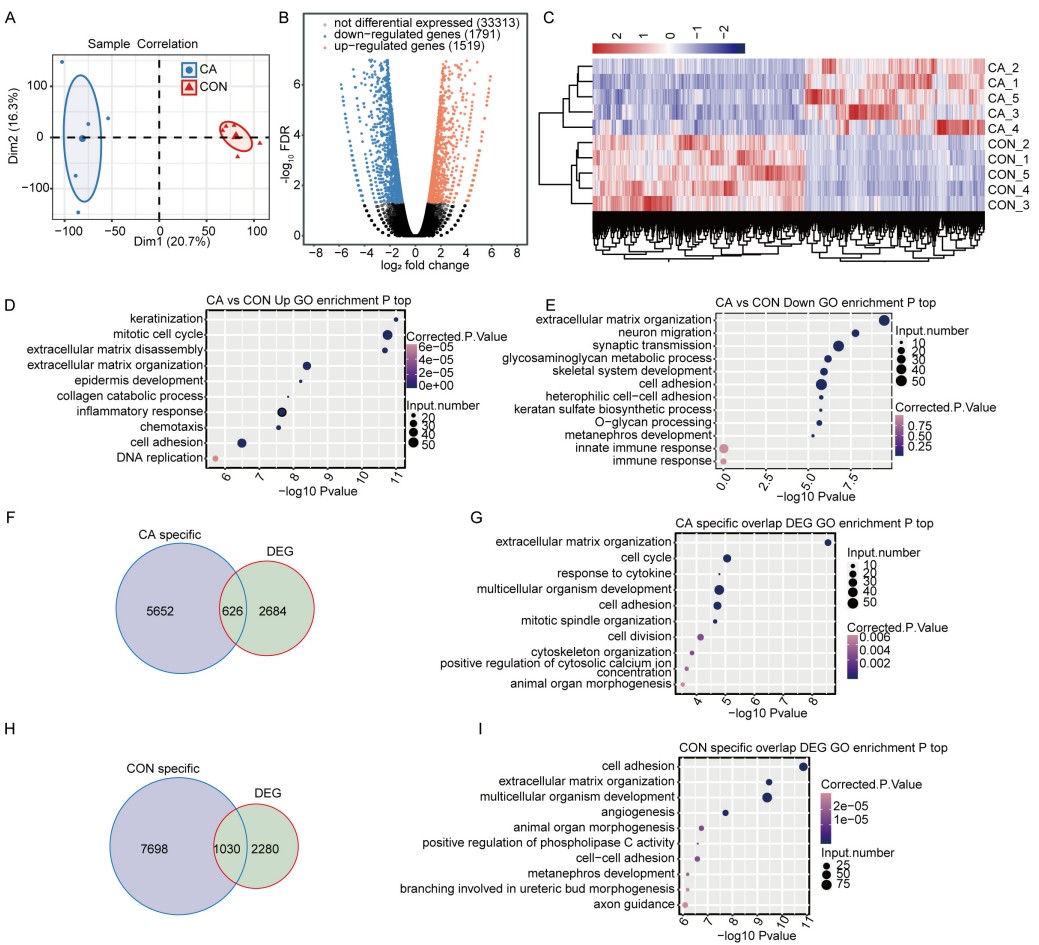

**Figure 3** M6A modification broadly changes gene expression in condyloma acuminatum (CA) and paired adjacent normal tissues (CON). (A) Principal component analysis (PCA) showing the expression pattern of all expressed genes in the two groups. Five biological replicates were prepared for each group, and the confidence ellipse for each group is shown as an ellipse with a blue or red color. (B) Volcano plot showing the number of DEGs in CA compared with CON samples with a cutoff of FDR <0.05 and FC ≥2 or ≤0.5. (C) Hierarchical clustering heatmap showing the expression pattern of DEGs among the samples. (D–E) Bubble plot showing the top 10 enriched GO terms (biological process) of up- or downregulated genes. (F) Venn diagram showing DEGs with CA specific m6A peaks. (G) Bubble plot showing the top 10 enriched GO terms (biological process) of DEGs with CA-specific m6A peaks. (H) Venn diagram showing DEGs with CON-specific m6A peaks. (I) Bubble plot showing the top 10 enriched GO terms (biological process) of DEGs with CON-specific m6A peaks.

DEGs in our study were RBPs (Fig. 4A). The RBP genes expressed at an expression level of FPKM >0.5 in at least one sample were obtained. A heatmap plot of these RBP genes showed that ten RBPs (TLR7, MATR3, RNASEK, RBP7, TST, RBM20, RNASE4, PDCD4, ZCCHC24, BICC1, NSUN7 and TLR3) had lower expression in CA samples than in CON samples (Fig. 4B). However, compared to CON samples, the expression of EMG1, RBM38, ENOX1, BRCA1, RNF17, RBP1, ZC3H12D, OASL, PCBP3, DQX1, DZIP1L, AZGP1, ADARB1, DNMT3B, ZC3H12C, DNMT1 and RNASEH2A increased in CA samples (Fig. 4B).

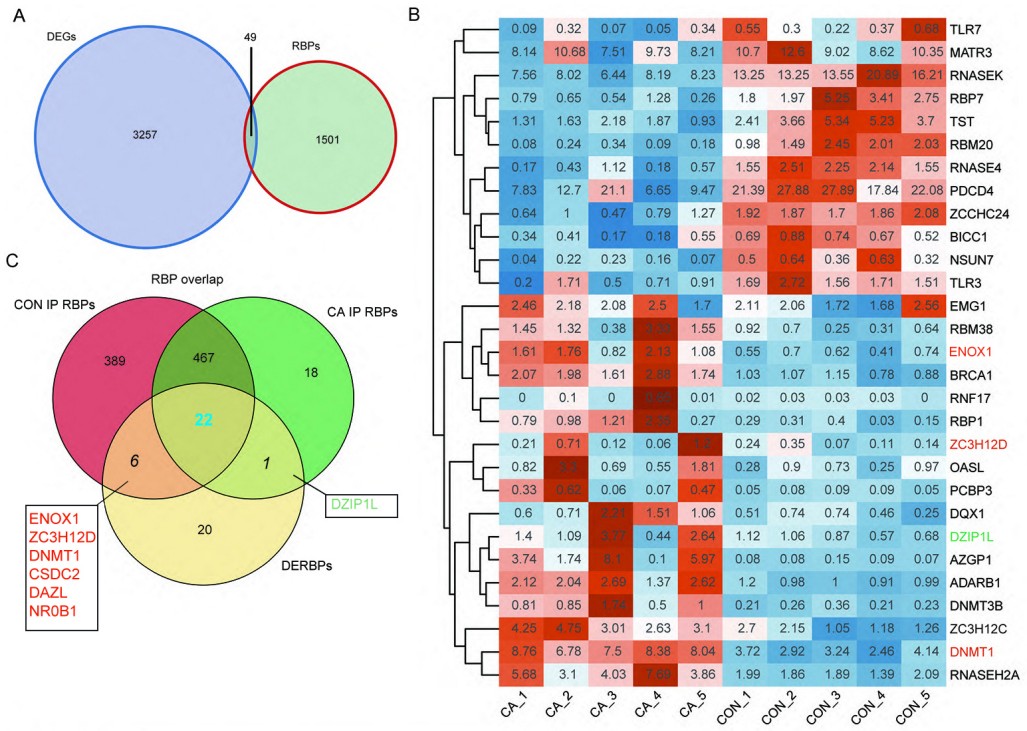

**Figure 4 M6A-dyregulated differentially expressed RNA binding proteins (RBPs) in condyloma acuminatum (CA) and the paired adjacent normal tissues (CON).** (A) Venn diagram showing differentially expressed RBPs. (B) Hierarchical clustering heatmap showing the expression levels of differentially expressed RBPs with the FPKM values >0.5 in at least one sample. (C) Venn diagram showing differentially expressed RBPs that identified m6A peaks in CA and CON samples.

Then, an overlap analysis was performed to identify the CON- or CA-specific m6A peak-associated differentially expressed RBP genes. The results showed that one differentially expressed RBP gene (DZIP1L) had a CA-specific m6A peak, and six differentially expressed RBP genes (ENOX1, ZC3H12D, DNMT1, CSDC2, DAZL, NR0B1) had a CON-specific m6A peak (Fig. 4C). DNMT1 and ZC3H12D had a m6A peak in CON samples (Figs. 5A–5B), while DZIP1L had a m6A peak in CA samples (Fig. 5C). Interestingly, all of these three RBP genes of DNMT1, ZC3H12D and DZIP1L showed higher expression in CA than CON samples (Figs. 5D–5F). To verify the expression of these three RBP genes with CON-specific or CA-specific m6A peaks, RT-qPCR was conducted to quantify their changes in additional CON and CA tissue samples in larger numbers. The RT-qRCR results revealed a significant increase in DNMT1, ZC3H12D and DZIP1L (Figs. 5G–5I) and a decrease in NR0B1, DAZL, ENOX1 and CSDS2 (Figs. S2A–S2D) in CA tissues compared to CON tissue, which was consistent with the RNA-seq data (Figs. 5D–5F, and Figs. S2E–S2H). These results suggested that m6A modification may be an important regulatory contributor to RBPs in CA tissues further influencing their protein expression.

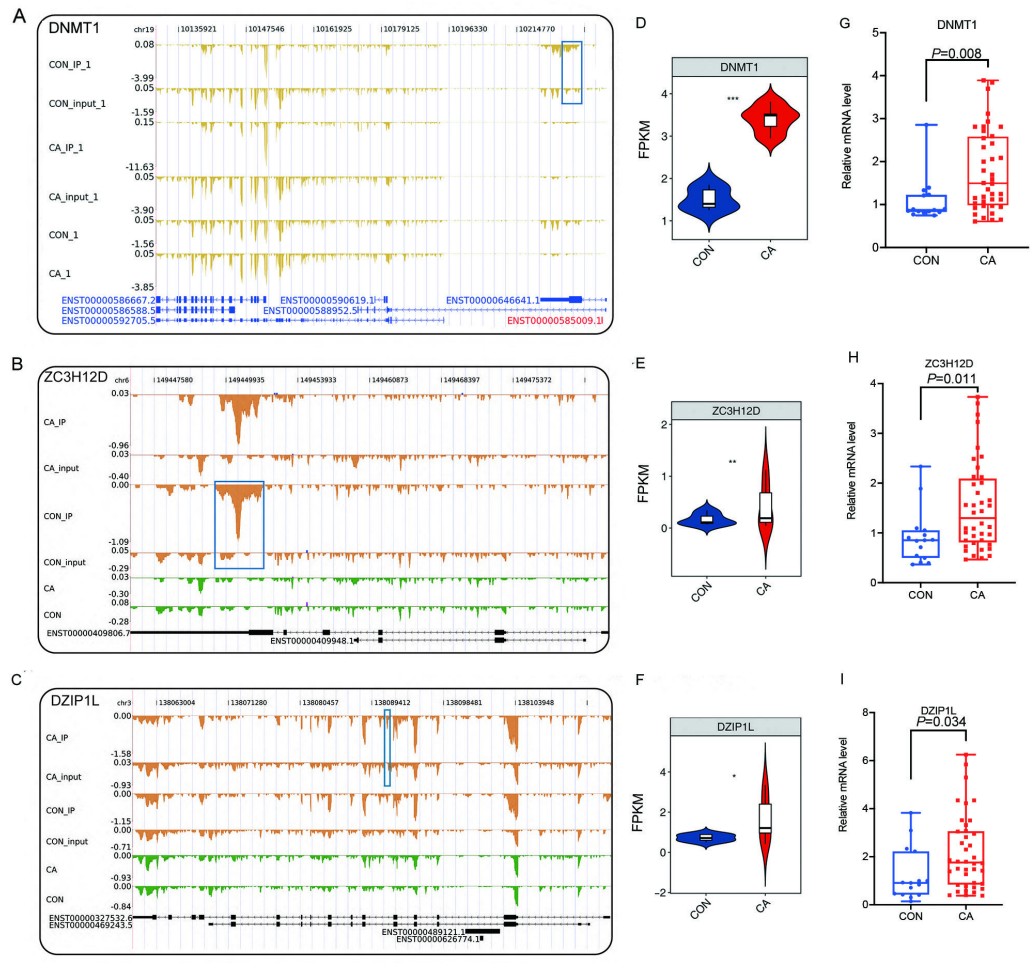

**Figure 5** **Reads density and expression of target genes.** (A–C) Read density showing the m6A signals of DMNT1, ZC3H12D and DZIP1L. Altered m6A peaks are indicated in the blue box. (D–F) Violin plot showing the FPKM expression of DMNT1, ZC3H12D and DZIP1L using RNA-sequencing. *, $P < 0.05$; **, $P < 0.01$; ***, $P < 0.001$. (G–I) Bar plot showing the relative expression of DMNT1, ZC3H12D and DZIP1L in additional CON and CA tissue samples using RT-qPCR.

## Differentially expressed RBPs with altered m6A methylation potentially regulate AS in condyloma acuminatum

RBPs also play vital roles in the regulation of RNA processing, including AS of both host cellular RNA and HPV RNA (*Fredericks et al., 2015*; *Nilsson, Wu & Schwartz, 2018*). We wondered whether aberrant expression of RBPs caused changes in the AS pattern of some genes in CA tissue. In our data, a total of 1,703 RASEs were identified with 96 intron-retention AS events (IR) and 1,607 other types of AS events (NIR) (Fig. 6A). Most of the NIR events were A5SS (498), A3SS (372), cassette Exon (257) and Exon Skipping (254) (Fig. 6A).

Then, the percentage spliced in (PSI) value of these RASEs was calculated for each sample. A heatmap of the unsupervised hierarchical clustering of the PSI values of these

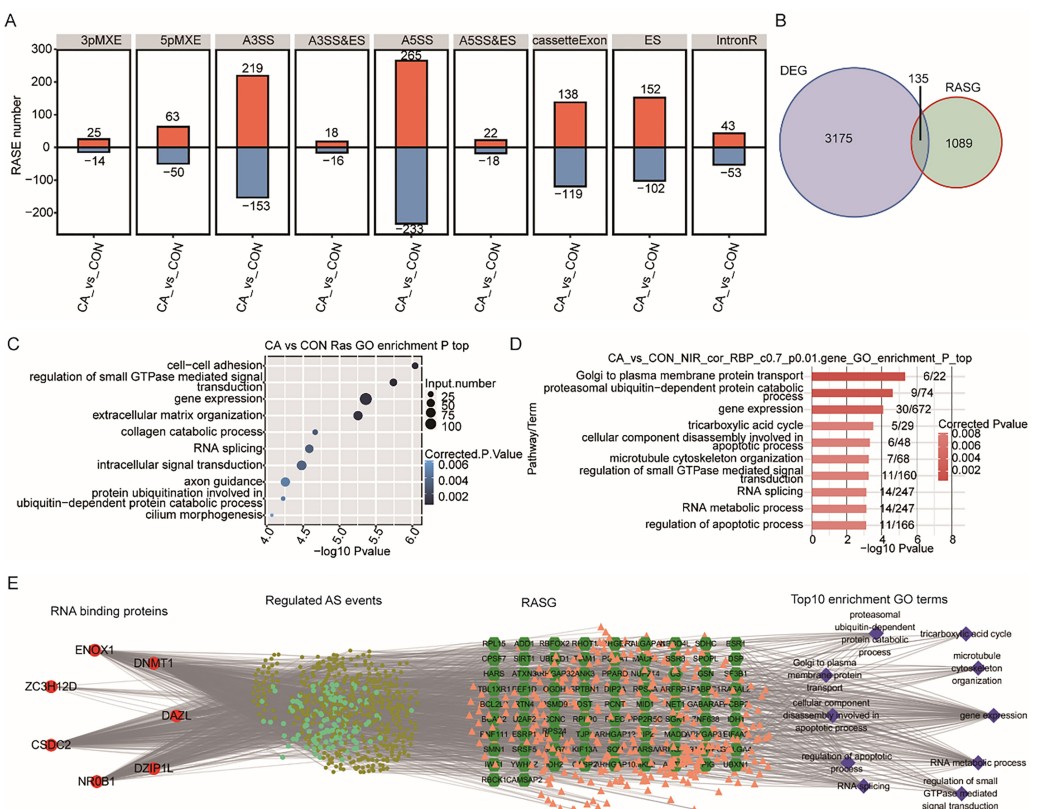

**Figure 6** **Correlation analysis between m6A-dyregulated differentially expressed RBPs and alternative splicing (AS) events.** (A) Bar plot showing the number of nine types of dysregulated alternative splicing events (RASEs) in CA samples compared with CON samples. (B) Venn diagram showing the overlapping genes between DEGs and RASGs. (C) Bubble plot showing the top 10 enriched GO terms (biological process) of RASGs. (D) Bar plot exhibiting the most enriched GO terms (biological process) of genes corresponding to RASEs coexpressed with m6A-modified samples-specific RBPs. (Correlation coefficient > 0.7 and $P$ value < 0.01). (E) The coexpression results between the seven m6A-modified samples-specific RBPs, RASEs, RASG and the most enriched biological processes (top 10).

RASEs showed that five CA samples and five CON samples could be clustered together (Fig. S3A). Genome location analysis showed that these RASEs occurred on a total of 1,224 genes (referring to RASGs). Overlap analysis of DEGs and RASGs showed that only 135 genes had changes in both expression and AS patterns between CA and CON samples (Fig. 6B). GO term analysis was conducted to explore the potential function of RASGs and showed that RASGs were enriched in the terms of cell–cell adhesion and extracellular matrix organization (Fig. 6C). Kyoto Encyclopedia of Genes and Genomes (KEGG) analysis showed that RASGs were enriched in pathways including adherens junction, focal adhesion and NF-kappa B signaling pathways (Fig. S3B). These results indicated that HPV infection could change the AS pattern of host cellular RNA in CA tissue.

To explore whether these seven differentially expressed RBPs with CON-specific and CA-specific m6A peaks affect the AS in CON and CA samples, coexpression analysis was performed based on the expression values of RBPs and PSI values of RASEs in all the ten

input samples. The RASGs with RASEs coexpressed with these seven RBPs (DZIP1L, ENOX1, ZC3H12D, DNMT1, CSDC2, DAZL, NR0B1) were extracted (correlation coefficient >0.7 and *P* value <0.01). Then, GO and KEGG analyses were performed to explore the potential functions of these RASGs. The results showed that these RASGs were significantly enriched in GO terms including cellular component disassembly involved in apoptotic process and regulation of apoptotic process (Fig. 6D). KEGG analysis showed that these RASGs were enriched in pathways including regulation of autophagy, ECM-receptor interaction and adherens junction (Fig. S3C). Furthermore, a coexpression network among the seven RBPs and RASEs, RASGs, and the top enriched GO terms of RASGs was constructed, as shown in Fig. 6E.

Finally, we focused on the three RBPs of DNMT1, ZC3H12D and DZIP1L that had been confirmed using RT-qPCR and their probable regulatory roles on in RASEs and RASGs. Interestingly, the top enriched GO terms of RASGs that might be regulated by DMNT1 were also involved in the regulation of apoptotic processes and cellular component disassembly involved in apoptotic processes (Fig. 7A). Then, some genes in the coexpression network with changes in AS patterns were presented as differences in PSI values among CA and CON samples. Our results showed that PSI values of A5SS events on RASAL2 and YWHAZ were higher in the CON group than in the CA group (Figs. 7B–7C), which was confirmed in additional CON and CA tissue samples using RT-qPCR (Figs. 7D–7E). Notably, m6A peaks on the same site of these three genes were detected for both the CA and CON samples (Figs. 7B–7C), which indicated that m6A methylation did not affect AS. In addition, we analyzed the expression level of DMNT1 in different groups with various clinical characteristics, and found that the more DMNT1, the more times or duration of treatment (Figs. S4A–S4B). It was suggested that highly expressed DMNT1 indicated a greater likelihood of relapse in the course of CA treatment. Taken together, these results indicated that differentially expressed DMNT1 with CON-specific m6A peaks could result in changes in the AS pattern of some genes involved in terms of the regulation of apoptotic processes, thus participating in the pathological process of CA.

# DISCUSSION

The modification of m6A on mRNAs and lncRNAs plays a pivotal role in biological processes and disease pathogenesis. It has been reported that diverse viruses can induce alterations in transcriptome-wide m6A modification in host cellular RNA (*Lichinchi et al., 2016*; *Baumgarten et al., 2019*; *Li et al., 2021*) Although HR-HPV 16/18 infection increases the expression of IGF2BP2 to stabilize m6A methylated MYC in cervical cancer (*Hu et al., 2022*), and high levels of METTL3 in HR-HPV-associated cancer mediate an immunosuppressive tumor microenvironment (*Yu et al., 2022*), there are no data about the effect of LR-HPV infection on the transcriptome-wide m6A distribution on cellular RNA. For the first time, we reported the m6A methylation profiles of cellular mRNAs in CA tissues. In total, an obvious differential level and distribution of m6A methylation was observed in CA tissues compared with paired normal tissues. We further focused on the DEGs with CA- or CON-specific m6A and their functional enrichment. Based

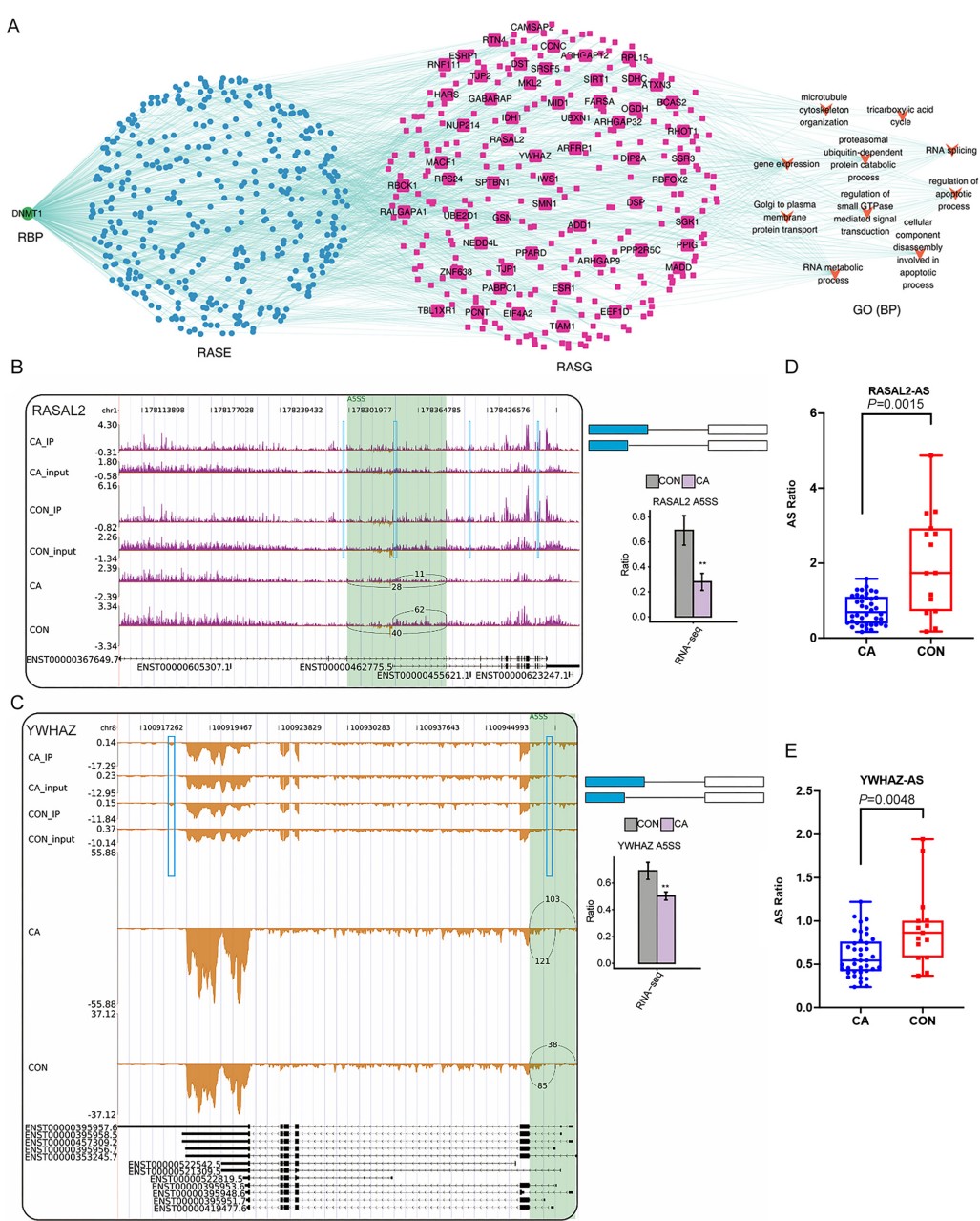

**Figure 7** **Correlation analysis between DNMT1 and alternative splicing (AS) events.** (A) The coexpression results between DNMT1, RASEs, RASGs and the most enriched biological processes (top 10). (B–C) Read density and bar plot showing the AS events and m6A signals in the RASAL2 and YWHAZ genes. **, *P* < 0.01. (D–E) AS ratio of RASAL2 and YWHAZ genes in additional CON and CA tissue samples using RT–qPCR.

on the bioinformatic analysis, we found that DEGs with either CA-specific or CON-specific m6A were mainly enriched in the terms cell cycle, response to cytokine and cell adhesion, angiogenesis and regulation of apoptotic process, which are closely related to

the pathogenesis of CA. Therefore, we inferred that HPV initiated the occurrence and development of CA by changing the m6A modification in host cellular RNA.

RBPs play vital roles in maintaining the transcriptome by controlling the processing and transportation of RNA at the posttranscriptional level , such as RNA stability, AS, translation, modification, and localization (*Chen et al., 2019*). On the one hand, RBPs such as eIF3 and IGF2BPs act as m6A readers (*Lee et al., 2016*; *Huang et al., 2018*). On the other hand, nonreader RBPs are critical in RNA biology, including AS which is involved in maintaining protein diversity and mRNA stability (*Qin et al., 2020*). Thus, we focused on RBPs and identified seven differentially expressed RBP genes (ENOX1, ZC3H12D, DNMT1, CSDC2, DAZL, NR0B1 and DZIP1L) with altered m6A methylation. To explore the roles of these RBPs in AS, we further analyzed the differential RASEs coexpressed with the seven RBPs in the same CA samples and found that RASGs were enriched in GO terms of cell −cell adhesion and extracellular matrix organization and pathways of adherens junction, focal adhesion and NF-kappa B signaling. To the best to our knowledge, few studies have comprehensively characterized the global modifications of RNA splicing signatures in CA. More interestingly and surprisingly, the RASGs with RASEs coexpressed with these seven RBPs were significantly enriched in GO terms, including cellular component disassembly involved in apoptotic process and regulation of apoptotic process, and pathways including regulation of autophagy, ECM-receptor interaction and adherens junction. We inferred that HPV infection regulated the expression of these RBPs by changing m6A methylation, and these differentially expressed RBPs would affect AS in CA tissue to regulate the processes of apoptosis, autophagy and cellular adhesion in CA.

We further constructed a network coexpression relationship network between RASEs and RBP genes of DNMT1, ZC3H12D and DZIP1L, which was validated in a larger number of samples. We found that the ASEs of RASAL2 and YWHAZ and the RBP gene DNMT1 were closely associated in the coexpression network, and involved in the regulation of apoptotic processes, which indicated that DNMT1 may contribute to CA progression. DNMT1, DNA methyltransferase 1, is a core enzyme that maintains intracellular DNA methylation (*Wong, 2021*) and functions as an RBP (*Unterberger, Torrisani & Szyf, 2008*; *Xu et al., 2018*). It was reported that lncRNA SAMD12-AS1 interacted with DNMT1 to inhibit the P53 signaling pathway and further promoted gastric cancer progression (*Lu et al., 2021a*). The lncRNA MIR210HG promotes the proliferation and invasion of non-small cell lung cancer by binding to DNMT1 directly, thereafter upregulating methylation of the CACNA2D2 promoter (*Kang et al., 2019*). Inactivation of DNMT1/3b in colon carcinoma cells induced a partial epithelial-mesenchymal transition associated with increased CD44 variant exon skipping (*Batsche et al., 2021*). There are two novel variants of lncRNA LINC00887 in tongue squamous carcinoma, namely, LINC00887_TSCC_short (887S) and LINC00887_TSCC_long (887L). 887L activated carbonic anhydrase IX transcription by recruiting HIF1 $\alpha$, while 887S suppressed carbonic anhydrase IX through DNMT1-mediated DNA methylation (*Shen et al., 2021*). Herein, we confirmed that two apoptosis-related genes, RASAL2 (*Koh et al., 2021*; *Xiong et al., 2021*) and YWHAZ (*Nishimura et al., 2013*; *Zhang et al., 2021*) were associated with 5ASS ASEs in both our RNA-sequencing data and RT-qPCR validation data. However, future functional experiments should be

conducted to identify whether DNMT1 funcions as a nonreader RBP and affects AS of downstream genes, especially RASAL2 or YWHAZ, in HPV-infected CA tissue.

Admittedly, there are several limitations in our present study. (1) The sample size of five pairs of CA and their adjacent control tissues is relatively small. More self-controlled CA samples should be included for verification in the future. (2) M6A methylation level of these screened target RBPs should be validated in a larger number of CA samples using MeRIP −qPCR or m6A sequencing. (3) Whether or not the expression of DNMT1 is regulated by m6A methylation, whether the differential m6A status of DNMT1 affects the AS of its downstream genes, the gain- and loss-functional experiments *in vitro* should be performed in the future.

## CONCLUSIONS

Our study revealed that the profiles of m6A methylation, RBPs and RASEs in CA patients differed from those in normal controls. The analyses of the GO and KEGG pathways showed that these differentially expressed RBPs with altered m6A methylation might play certain roles in the pathogenesis of CA, especially in the regulation of apoptotic processes. Mechanistically, these RBPs played their roles might be *via* changing the AS of target genes in CA. Our study provides basic data and hints for the further exploration of the roles of m6A and the discovery of novel therapeutic targets in CA.

### Funding

This research was funded by the Natural Science Foundation of Zhejiang Province (LY22H110001) and the Medical Science and Technology Project of Zhejiang Province (2020KY913). The funders had no role in study design, data collection and analysis, decision to publish, or preparation of the manuscript.

### Grant Disclosures

The following grant information was disclosed by the authors:
The Natural Science Foundation of Zhejiang Province: LY22H110001.
the Medical Science and Technology Project of Zhejiang Province: 2020KY913.

### Competing Interests

The authors declare there are no competing interests.

### Author Contributions

- Xiaoyan Liu conceived and designed the experiments, analyzed the data, authored or reviewed drafts of the article, and approved the final draft.
- Bo Xie performed the experiments, analyzed the data, authored or reviewed drafts of the article, and approved the final draft.
- Su Wang performed the experiments, analyzed the data, prepared figures and/or tables, and approved the final draft.

- Yinhua Wu analyzed the data, prepared figures and/or tables, and approved the final draft.
- Yu Zhang performed the experiments, prepared figures and/or tables, and approved the final draft.
- Liming Ruan conceived and designed the experiments, authored or reviewed drafts of the article, and approved the final draft.

## Human Ethics

The following information was supplied relating to ethical approvals (*i.e.*, approving body and any reference numbers):

The research was approved by The Institutional Review Board of the First Affiliated Hospital, Zhejiang University School of Medicine. (2022-795)

## Data Availability

The data is available at GEO: GSE172140.

## Supplemental Information

Supplemental information for this article can be found online at http://dx.doi.org/10.7717/peerj.17376#supplemental-information.

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
