# Peer review of "Alteration of RNA m6A methylation mediates aberrant RNA binding protein expression and alternative splicing in condyloma acuminatum"

_PeerJ, doi:10.7717/peerj.17376_

## Round 0.1 · original submission · Major Revisions

Please address the concerns of all reviewers and amend the manuscript accordingly.

Reviewer 1 has suggested that you cite specific references. You are welcome to add it/them if you believe they are relevant. However, you are not required to include these citations, and if you do not include them, this will not influence my decision.

Reviewer 1 ·

Basic reporting

no comment

Experimental design

no comment

Validity of the findings

no comment

Additional comments

How was hpv genotyping performed in condyloma acuminata itself?
Only those with hpv type 6/11 were included.
How many types of HPV were investigated in the study?
Low risk hpv types are responsible for CA. But it is not the rule alone. High risk HPV types can also be found with CA.
The authors can add the study below as a reference to increase the value of content.
Sarier M, Sepin N, Emek M, et al. Evaluation of the optimal sampling approach for HPV genotyping in circumcised heterosexual men with genital warts. J Infect Chemother. Published online January 2023. doi:10.1016/J.JIAC.2023.01.017

Reviewer 2 ·

Basic reporting

The authors have performed an extensive bioinformatics analysis of m6A methylation, gene expression, and alternative splicing changes in human condyloma acuminatum tissues compared to normal controls. They propose the model that HPV-induced changes in m6A methylation lead to RBP dysregulation, which alters splicing of downstream gene targets involved in CA pathogenesis.
Overall the study is generally clear and the design and findings appear novel and meaningful. The introduction provides adequate background information and the literature is well referenced and relevant. The structure of the article, including figures, tables, and the sharing of raw data, adheres to professional standards, contributing to the overall clarity and comprehensiveness of the work.

However, the authors should address issues to strengthen the quality before the paper is considered for publication:
1. The sample size of 5 CA and 5 control samples for the initial sequencing experiments is relatively small. Authors should increase the number of samples.

2. While the authors validated their sequencing results by incorporating additional samples and employing the RT-qPCR method, it is noteworthy that these samples were not utilized to validate crucial findings, such as alterations in m6A methylation and changes in alternative splicing.

3. Provide more specifics on the human tissue samples - cancer type/stage, treatment status etc.

4. Some conclusions about the exact regulatory relationships between the m6A marks, RBPs, and splicing events are overstated without additional functional validations.

Experimental design

As I mentioned before, the authors have tried to validate sequencing results by analyzing additional samples (40 CA specimens and 15 HPV-negative vaginal mucosal tissues) with RT-qPCR for validation, but they are not paired as the sequencing samples (tumor and adjacent mucosal tissue), and the results are underpowered.

Validity of the findings

no comment

Reviewer 3 ·

Basic reporting

The article uses clear English but could benefit from a smoother transition between discussing HR-HPV and m6A modification in CA. Expand on current research to contextualize your study better. Ensure all figures are clearly labeled to avoid overlap and improve readability. Check for typographical errors (e.g., line 105) and ensure all references are correctly cited and found in the reference section (line 134).

Experimental design

-The research question is well-defined, but the clarity and logical flow of the procedural steps need improvement.

-Ensure there is more clarity regarding the normalization methods, with a shift from mentioning RPKM in the Methods section to FPKM in the Results section.

-Throughout the manuscript, maintain consistency in mentioning the tools used, and clearly specify the parameters considered, addressing any disparities between the Methods and Results sections (e.g., DESEQ).

-Figures appear congested, and the values marked on them overlap, requiring adjustments for better readability.

-Include essential details about figure color groups, notations, and p-values. For instance, clarify what is meant by "upregulated" and "downregulated" in Figure 2B.

-In the Results section, provide clearer information about the methylated samples and their relevance.

-Ensure consistency in referencing figures throughout the manuscript (e.g., line 251, 305).

-Clarify the details of the ABLIRC pipeline, including the algorithm used and the process of motif detection. The manuscript should provide a clear and consistent explanation of the pipeline details.

-When referring to PCA on line 192 for unique reads in each gene, specify whether the figure described pertains to genes or samples for better comprehension.

-Improve the clarity of Figure 1B by separating and labeling peaks corresponding to the 1st and 2nd scales, avoiding overlap.

-Address the discrepancy in values mentioned on line 202 compared to the figure, ensuring consistency.

-Enhance figure clarity by highlighting motifs for better understanding.

-Consider presenting the table format for the 49 DEGs or explicitly mention the genes not represented on the heatmap in Figure 3B, as these genes are frequently discussed in the text.

-Review figures, legends, titles, and descriptions to ensure they provide relevant and accurate information.

Validity of the findings

The data presented demonstrate robustness and statistical reliability, with effective control measures in place. However, the manuscript could benefit from improvements in data representation, sentence structure, and overall clarity. Additionally, there is an opportunity to delve deeper into the potential implications and broader significance of the findings within the field, aiming for a more coherent presentation.

Additional comments

In general, the manuscript maintains a well-structured format and offers a valuable contribution. To enhance clarity, consider implementing the aforementioned suggestions and reinforcing the connections between sentences for improved cohesiveness.

---

## Round 0.2 · Minor Revisions

Please address the remaining issues pointed out by reviewer #1 and amend the manuscript accordingly.

Reviewer 1 ·

Basic reporting

ok

Experimental design

ok

Validity of the findings

ok

Additional comments

comments are addressed.

Reviewer 2 ·

Basic reporting

The author addressed all my inquires, and I believe the manuscript is ready for publication acceptance.

Experimental design

no comment

Validity of the findings

no comment

Additional comments

no comment

Reviewer 3 ·

Basic reporting

1. Please review the workflow figure where "DESeq" is mentioned; this should be updated to "EdgeR" according to the response letter.
2. There are multiple typos throughout the text that need correction.
3. The description of the m6A-seq data analysis could be simplified.

Experimental design

The manuscript has been updated to address the issues raised in the previous comments.

Validity of the findings

The manuscript has been updated to address the issues raised in the previous comments.

Additional comments

N/A

---

## Round 0.3 · accepted · Accept

Thank you for addressing remining concerns. Revised manuscript is acceptable now.